# Multi-functional anodes boost the transient power and durability of proton exchange membrane fuel cells

Gurong Shen[1,10], Jing Liu [2,10], Hao Bin Wu [3,10], Pengcheng Xu[2], Fang Liu[2], Chasen Tongsh[4], Kui Jiao[4], Jinlai Li[5], Meilin Liu [6], Mei Cai [7], John P. Lemmon[8], Grigorii Soloveichik[8], Hexing Li[9], Jian Zhu[9✉] & Yunfeng Lu[2✉]

Proton exchange membrane fuel cells have been regarded as the most promising candidate for fuel cell vehicles and tools. Their broader adaption, however, has been impeded by cost and lifetime. By integrating a thin layer of tungsten oxide within the anode, which serves as a rapid-response hydrogen reservoir, oxygen scavenger, sensor for power demand, and regulator for hydrogen-disassociation reaction, we herein report proton exchange membrane fuel cells with significantly enhanced power performance for transient operation and low humidified conditions, as well as improved durability against adverse operating conditions. Meanwhile, the enhanced power performance minimizes the use of auxiliary energy-storage systems and reduces costs. Scale fabrication of such devices can be readily achieved based on the current fabrication techniques with negligible extra expense. This work provides proton exchange membrane fuel cells with enhanced power performance, improved durability, prolonged lifetime, and reduced cost for automotive and other applications.

[1] School of Materials Science and Engineering, Tianjin University, Tianjin 300350, P.R. China. [2] Chemical and Biomolecular Engineering, University of California, Los Angeles, CA 90095, USA. [3] School of Materials Science and Engineering, Zhejiang University, Hangzhou 310027, P.R. China. [4] State Key Laboratory of Engines, Tianjin University, Tianjin 300350, P.R. China. [5] State Key Lab of Coal-Based Low Carbon Energy, ENN Technology and Development Co. Ltd., Langfang, Hebei 065001, P.R. China. [6] School of Materials Science and Engineering, Georgia Institute of Technology, 771 Ferst Drive, Atlanta, GA 30332-0245, USA. [7] General Motors Research and Development Center, Warren, MI 48090-9055, USA. [8] Advanced Research Projects Agency-Energy, Washington, DC 20024, USA. [9] Key Laboratory of Resource Chemistry of Ministry of Education, Shanghai Key Laboratory of Rare Earth Functional Materials, College of Chemistry and Materials Science, Shanghai Normal University, Shanghai 200234, P.R. China. [10] These authors contributed equally: Gurong Shen, Jing Liu, Hao Bin Wu. ✉email: jianzhu@shnu.edu.cn; luucla@ucla.edu

Fuel cells are considered to constitute a promising technology for efficient and clean power generation in the twenty-first century. Proton exchange membrane fuel cells (PEMFCs), in particular, with low operation temperature, high-power density, fast start-up ability and suitability for discontinuous operation, have been regarded as the most promising candidate for auto-motive applications[1–4]. The power output of PEMFCs is generally rated under steady-state operation. In reality, however, PEMFCs are frequently operated under transient states, during which power-output delay often occurs, limiting power performance[5,6]. Large-size fuel cells with higher rated power or auxiliary energy-storage systems could be utilized to mitigate this limitation, but unavoidably increases costs[7,8]. Furthermore, operating PEMFCs under transient states (e.g., start-up, shutdown, and high-power loading) dramatically shortens their lifetime[9–11]. For example, PEMFCs may exhibit a lifetime of over 30,000 h under steady-state operation; however, an average lifetime (time lasting until a 10% voltage drop is reached) of 3900 h was projected under a simulated road-testing condition. Despite decades of effort, broad adaption of PEMFCs is still impeded by cost and lifetime[3,11].

PEMFCs rely on continuous electrochemical reactions of hydrogen and oxygen, which involve fuel transport, disassociation of hydrogen into electrons and protons, transport of electrons and protons, and their subsequent reactions with oxygen in cathodes that produce water and generate power. The transport of fuel is related to cell architecture, fuel-flow control, water and heat management, and other factors[5,12,13]. Fuel starvation often occurs during transient operation, which not only deteriorates power output, but also shortens the lifetime[14–17]. Even with a sufficient supply of hydrogen, a delayed supply of protons still occurs upon high-power demand, resulting in transient polar-ization that causes power-output delay lasting up to tens of sec-onds[17]. In addition, during the start-up process, oxygen diffused from cathodes to anodes increases anode potential and triggers degradation of cathodes, further shortening the lifetime of PEMFCs[18–20]. Although many strategies have explored to miti-gate the impacts associated with fuel starvation and start-up/shutdown, such as using catalyst supports with improved corro-sion resistance, implementing a water oxidation electrocatalyst in the anode, and adapting extra control systems[21–23], there is still lack of effective yet low-cost solutions for these limitations.

We envision that PEMFCs with significantly enhanced tran-sient performance and prolonged lifetime could be made by integrating anodes with a thin layer of tungsten oxide ($WO_3$)[24]. $WO_3$ has been extensively explored as co-catalysts and catalyst supports for fuel cells, which could promote the hydrogen-disassociation reaction and improve the tolerance of Pt for carbon monoxide (CO)[25,26]. Beyond the catalytic applications, $WO_3$ with a hexagonal crystalline structure, in particular, is a highly stable proton-electron mixed conductor, of which a high capacity of protons could be stored in a highly reversible and rapid manner at $-0.4$ to 0.5 V vs. reversible hydrogen electrode in an acidic environment[27,28]. These characteristics uniquely qualify its use as a rapid-response hydrogen reservoir, oxygen scavenger, sensor for power demand, and regulator for hydrogen-disassociation reaction.

Here we present our design of a hybrid PEMFC device as illustrated in Fig. 1, which adapts a traditional membrane-electrode assembly (MEA) architecture that comprises two gas diffusion layers (GDLs), two Pt/carbon catalyst layers, and a proton-exchange membrane. In the anode, a thin layer of hex-agonal $WO_3$ is integrated between the GDL and the Pt/carbon catalyst layer. During a normal operation, $H_2$ passes through the GDL and the $WO_3$ layer, and is converted to protons and elec-trons in the catalyst layer (reaction i). The protons and electrons are then transported crossing the proton-exchange membrane and

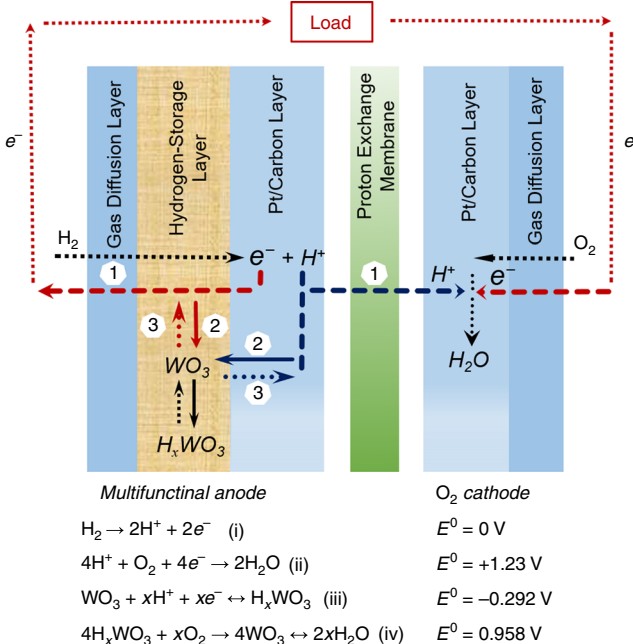

**Fig. 1 Design of a hybrid proton-exchange membrane fuel cell.** Schematic of a membrane-electrode assembly (MEA) for the hybrid PEMFC with a $WO_3$-based multifunctional anode. The cell functions through the reactions (i) and (ii), where electrons and protons go through the pathway 1. The $WO_3$ layer serves as a rapid-response hydrogen reservoir storing and releasing electrons and protons based on the reaction (iii) through pathways 2 and 3, respectively. The $WO_3$ layer also serves as a scavenger for oxygen invaded to the anodes through the reaction (iv) and a regulator for the hydrogen-disassociation reaction (i).

external circuit, respectively, reacting with $O_2$ in the cathode in a manner similar to that of the traditional PEMFCs (pathway 1, reaction ii). As-generated protons and electrons also react with the $WO_3$ and are stored in the form of $H_xWO_3$ (pathway 2, reaction iii). In response to fuel starvation or high-power demand, $H_xWO_3$ is converted back to $WO_3$, releasing electrons and protons to compensate the fuel shortage (dash lines, pathway 3). The $WO_3$ layer is then recharged spontaneously through the reaction iii, reloading with electrons and protons for the next compensating cycle. In addition, $H_xWO_3$ also reacts with $O_2$ diffused to the anode, forming $WO_3$ and $H_2O$ (reaction iv)[29,30]. Such $O_2$ sca-venger stabilizes anode potential, protecting the cathodes from degradation and prolonging the lifetime of PEMFCs. Moreover, hydrogen-disassociation reaction (i) occurs with low activation energy. As described by the Butler–Volmer equation, the rate of disassociation increases proportionally to (at low overvoltage) or exponentially with the anode overvoltage (at high overvoltage)[31]. In response to power demand, discharge of the $WO_3$ layer increases anode potential and promotes the disassociation reac-tion, providing more protons to avoid the power-output delay. In this context, the $WO_3$ layer not only serves as a rapid-response hydrogen reservoir and oxygen scavenger, but also constitutes a regulator for hydrogen-disassociation reaction, dynamically modulating anode overvoltage and disassociation of hydrogen according to power demand. Thus, we improve the transient power performance and durability of PEMFCs by integrating a $WO_3$ layer within the anode.

## Results
### Synthesis of $WO_3$/CNTs composites with rapid hydrogen storage-release capability.
To synthesize the $WO_3$, a hydrothermal

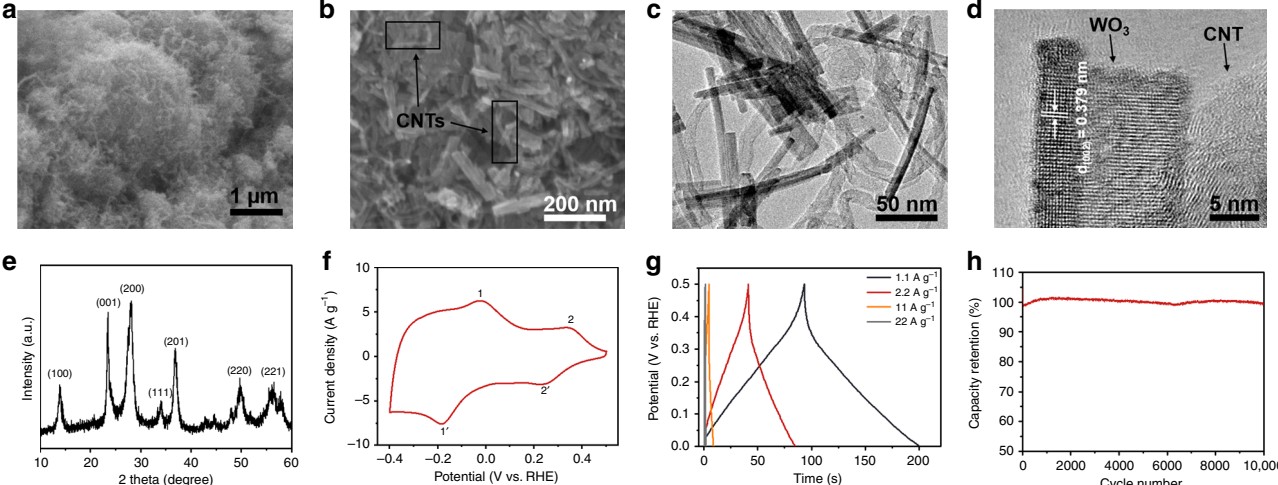

**Fig. 2 Structure and electrochemical characterizations of the composite. a, b** SEM, **c** TEM, and **d** high-resolution TEM images of the composite. **e** XRD pattern of the composite showing a hexagonal $WO_3$ crystalline structure. **f** A cyclic voltammetry curve of the composite from −0.4 to 0.5 V vs. reversible hydrogen electrode (RHE). **g** Galvanostatic charge/discharge curves of the composite at different current densities. **h** Cycling stability of the composite electrode in 0.5 M $H_2SO_4$.

reaction was conducted using sodium tungstate and ammonium sulfate as precursors at 180 °C[27,32]. To improve electronic conductivity, an appropriate amount of carbon nanotubes (CNTs) was dispersed within the precursor solution to form $WO_3$/CNTs composite. As shown in Supplementary Fig. 1, thermal gravimetric analysis (TGA) suggests that the $WO_3$ synthesized contains ~5.2 wt% of water. After subtracting the surface-adsorbed water, the $WO_3$ contains ~3.8 wt% of water within the crystals, corresponding to a molecular formula of $WO_3 \cdot (H_2O)_{0.5}$[27]. These water molecules are present as single-file water chains within the crystals, serving as biomimetic proton channels for fast proton conduction. Estimated from the weight lost between 450 and 600 °C due to combustion of the CNTs, the composite contains ~17 wt% of CNTs.

Figure 2a shows a representative scanning electron microscopic (SEM) image of the $WO_3$/CNTs composite, exhibiting a micrometer-sized particulate morphology. The magnified SEM image in Fig. 2b reveals that such particles are formed by entangled networks of $WO_3$ nanorods and CNTs. Transmission electron microscopic (TEM) image in Fig. 2c shows the detailed structure of the composite. The $WO_3$ nanorods are 5–10 nm in diameter and 60–100 nm in length, intertwining with CNTs of ~20 nm in diameter and up to several micrometers in length. High-resolution TEM image (Fig. 2d) displays a $WO_3$ nanorod with (002) lattice planes, which is intimately contacted with a multi-wall CNT. The phase purity of the $WO_3$/CNTs composite is confirmed by the X-ray diffraction (XRD) pattern in Fig. 2e. The diffraction peaks at 13.9, 23.5, 28.1, 34.0, 36.9, 46.7, and 56.4° are in concordance with the (100), (001), (200), (111), (201), (220), and (221) lattice planes of the hexagonal $WO_3$ (JCPDS #33-1387), respectively. The absence of CNTs diffraction peaks is likely due to its relatively low content. Our previous work suggests that such hexagonal $WO_3$ crystals can provide proton conductivity as high as 3.7 mS cm$^{-1}$ at 60 °C[27]. Such intertwining networks provide effective transport pathways for electrons and protons, endowing the composites with fast-response capability.

Electrochemical performance of the composite was first investigated in 0.5 M $H_2SO_4$ using a three-electrode cell, in which Pt foil and Ag/AgCl electrode were used as the counter and reference electrodes, respectively. Charge-storage behavior of the composite was measured by cyclic voltammetry (CV) from −0.4 to 0.5 V vs. reversible hydrogen electrode (RHE). As shown in

Fig. 2f, two pairs of well-defined redox peaks (the peaks at 0.23 and 0.34 V, and the peaks at −0.18 and −0.02 V, respectively) in Fig. 2f account for adsorption of protons at surface sites and reversible insertion of protons in channels of $WO_3$, respectively[27,28]. The symmetric shape of the CV curve indicates an ideal capacitive behavior, in line with our previous study. Figure 2g further shows the galvanostatic charge/discharge curves of the electrode at different current densities from 0.0 to 0.5 V vs. RHE; these linear curves further confirm a pseudocapacitive behavior. The electrode could deliver a capacity of 30.1 mAh g$^{-1}$ at current density 1.1 A g$^{-1}$ in this voltage window (Supplementary Fig. 2). Based on the theoretical capacity of $WO_3$ (~109 mAh g$^{-1}$, based on one-electron reaction), it is estimated ~1/3 of the $WO_3$ were utilized in this voltage window. At a high current density of 22 A g$^{-1}$, the electrode can still provide a reasonably high capacity of 5.4 mAh g$^{-1}$. It is also worth noting that such composite is electrochemically stable in an acidic condition, exhibiting negligible capacity fading over 10,000 charging–discharging cycles under a constant current of 11 A g$^{-1}$ (Fig. 2h). Such a long cycling life matches the lifespan of PEMFCs.

To evaluate their use as fast-response hydrogen reservoirs in PEMFCs, cells were assembled with a $WO_3$/CNTs electrode, a proton-exchange membrane, and a Pt/C electrode (Supplementary Fig. 3). Upon connecting the Pt/C electrode to hydrogen flow, the cell shows an open circuit voltage (OCV) of 0.6 V, corresponding to a potential of 0.6 V for the $WO_3$/CNTs electrode. The cell was then charged to 0.0 V using a constant current of 0.2 A g$^{-1}$, during which protons and electrons were inserted to the $WO_3$/CNTs electrode reaching a potential of 0.0 V vs. the dynamic hydrogen electrode (DHE). The insertion process exhibits a quasi-linear voltage-capacity curve with a capacity of ~33.6 mAh g$^{-1}$ (Supplementary Fig. 4). The hydrogen flow was then switched to oxygen flow, resulting in an OCV of ~1.0 V. Figure 3a shows the galvanostatic discharging curves of the cell, which could provide a discharging time of ~140 and 55 s under a discharging current density of 17.8 and 44.6 mA cm$^{-2}$, respectively. Based on the mass loading of $WO_3$ (~16.2 mg cm$^{-2}$), the electrode delivers a specific capacity of 33.2 mAh g$^{-1}$ at a current density of 1.1 A g$^{-1}$ (areal current density of 17.82 mA cm$^{-2}$) with a Columbic efficiency of ~100%. Even at a high current density of 22 A g$^{-1}$ (areal current density of 356.4 mA cm$^{-2}$), the

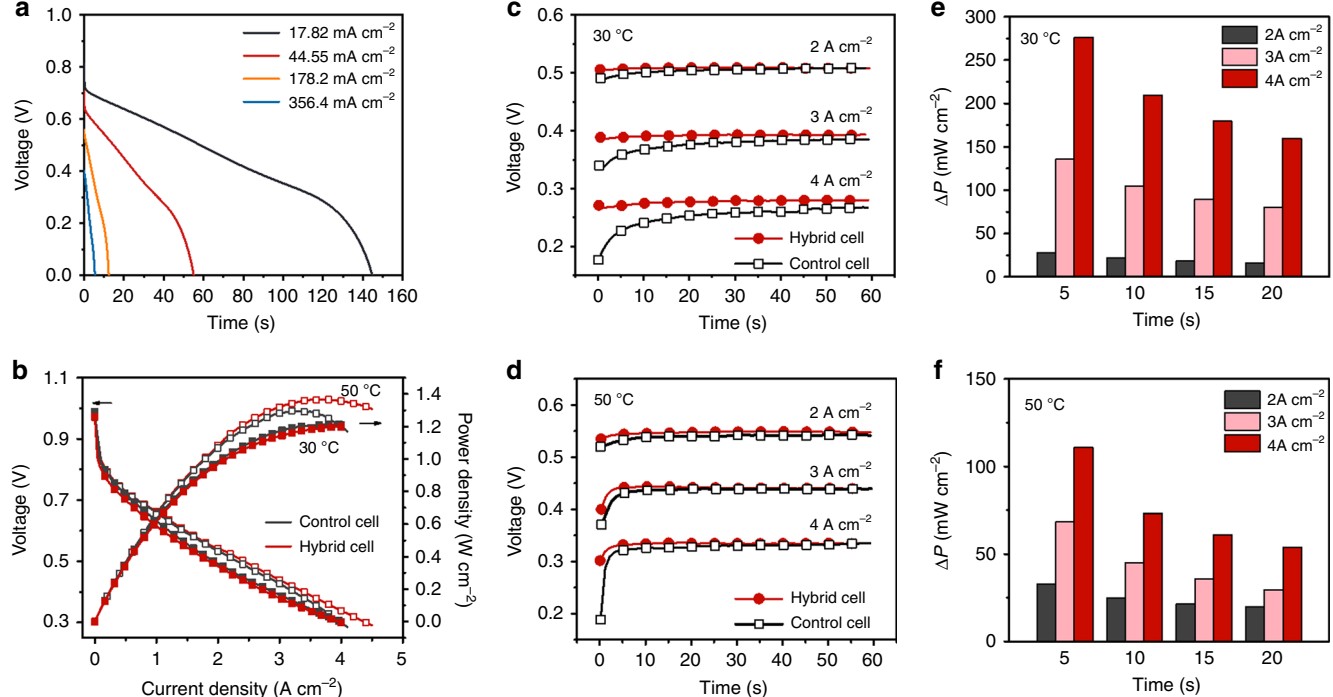

**Fig. 3 Discharging profiles of the composite electrode and transient performance of cells. a** Galvanostatic discharge curves of a $WO_3$ electrode discharged at different current densities measured in the air-capacitor configuration. **b** Polarization curves and power density of a hybrid cell (with $WO_3$) and a control cell (without $WO_3$) at 30 °C. Hundred percent humidified $H_2$ (stoichiometry = 1.5) and $O_2$ (stoichiometry = 4) were fed to the anodes and cathodes, respectively. **c, d** Voltages and **e, f** average $P$ at a time scale of 5, 10, 15, and 20 s of the control cell and the hybrid cell upon switching the current output from 0.05 A cm$^{-2}$ to different current outputs at 30 and 50 °C.

electrode can still deliver a high capacity of 25.6 mAh g$^{-1}$ (Supplementary Fig. 5). The cell could provide an average power density of 0.53 W g$^{-1}$ for 145 s or 10.7 W g$^{-1}$ for 5 s. Supplementary Fig. 6 further presents the Ragone plot of the electrode, which can provide a specific energy of 16.3 Wh kg$^{-1}$ at a high specific power density of 10.7 kW kg$^{-1}$. In terms of areal power density, Supplementary Fig. 7 shows the areal power density vs. discharging time. At a low current density of 17.82 mA cm$^{-2}$, corresponding to a power output of ~12 mW cm$^{-2}$, the cell can continuously function for over 100 s. Increasing the current density could boost the power output to ~100 mW cm$^{-2}$, which is equal to ~10% of peak-power density for typical PEMFCs (~1000 mW cm$^{-2}$), for ~5–10 s.

**Fuel cells with significantly enhanced transient power performance.** Fuel cells were then assembled to examine their transient performance. Figure 3b shows the polarization curves of a hybrid cell (with $WO_3$ at a mass loading of ~5.1 mg cm$^{-2}$) and a control cell (without $WO_3$) at 30 and 50 °C, respectively. Both cells exhibit nearly overlapped polarization curves and a similar peak-power density, implying that incorporating the $WO_3$ layer does not significantly alter the transport characteristic of the cells. To compare their transient performance, the cells were operated under a current density of 0.2 A cm$^{-2}$ and subjected to current outputs of 2, 3, and 4 A cm$^{-2}$, respectively, during which the cells were returned to 0.2 A cm$^{-2}$ after each increasing-current test. Figure 3c shows their voltage–time profiles at 30 °C. For the control cell, voltage increases with time approaching a steady voltage, indicating a power-output delay that becomes more pronounced with increasing current output. For example, a voltage undershoot of ~100 mV is observed with the current output of 4 A cm$^{-2}$ (corresponding to 100% of the maximum power

output), which takes more than 30 s to reach the steady voltage. In contrast, the hybrid cell shows much less delay, indicating improved power performance. Consistently, both cells exhibit higher voltages at 50 °C due to improved reaction and transport kinetics, while the hybrid cell still shows significantly less voltage delay than the control cell (Fig. 3d).

Supplementary Fig. 8a compares their power-output differences ($\Delta P$), which are estimated by subtracting the power density of the control cell from that of the hybrid cell. Upon changing the current density from 0.05 to 4 A cm$^{-2}$ at 30 °C, $\Delta P$ reaches 378 mW cm$^{-2}$ at the beginning and decreases with time. The average $\Delta P$ within a transient period of 5, 10, 15, and 20 s is 276, 210, 179, and 160 mW cm$^{-2}$, corresponding to 23%, 17.5%, 15%, and 13% of the maximum power output, respectively (Fig. 3e). The energy-output difference ($\Delta E$) is approximately 1.38 and 2.68 J cm$^{-2}$ for the first 5 and 15 s transient period, respectively. Supplementary Fig. 8b shows the $\Delta P$ profiles at 50 °C, which are decayed more rapidly with time. This result is consistent with the faster reaction and transport kinetics. As a result, the average $\Delta P$ within the same transient period is less than that of 30 °C; nevertheless, $\Delta P$ at the transient period of 5 s is still equivalent to ~10% of the maximum power output (Fig. 3f).

These studies strongly suggest that the $WO_3$ layer does serve as rapid-response hydrogen reservoirs enhancing transient performance. However, based on the specific energy/power of the $WO_3$ composite, the maximal energy that it could provide is ~0.3 J cm$^{-2}$ at an average power output of 54 mW cm$^{-2}$ or 0.34 J cm$^{-2}$ at an average power output of 27 mW cm$^{-2}$ (Supplementary Fig. 7), which are noticeably smaller than the $P$ and $E$ obtained. To examine this discrepancy, an equivalent circuit was built, in which the voltage source is represented by $U_0 - V_{act}$, where $U_0$ is cell voltage and $V_{act}$ is activation polarization (Fig. 4a). The anode and cathode are represented

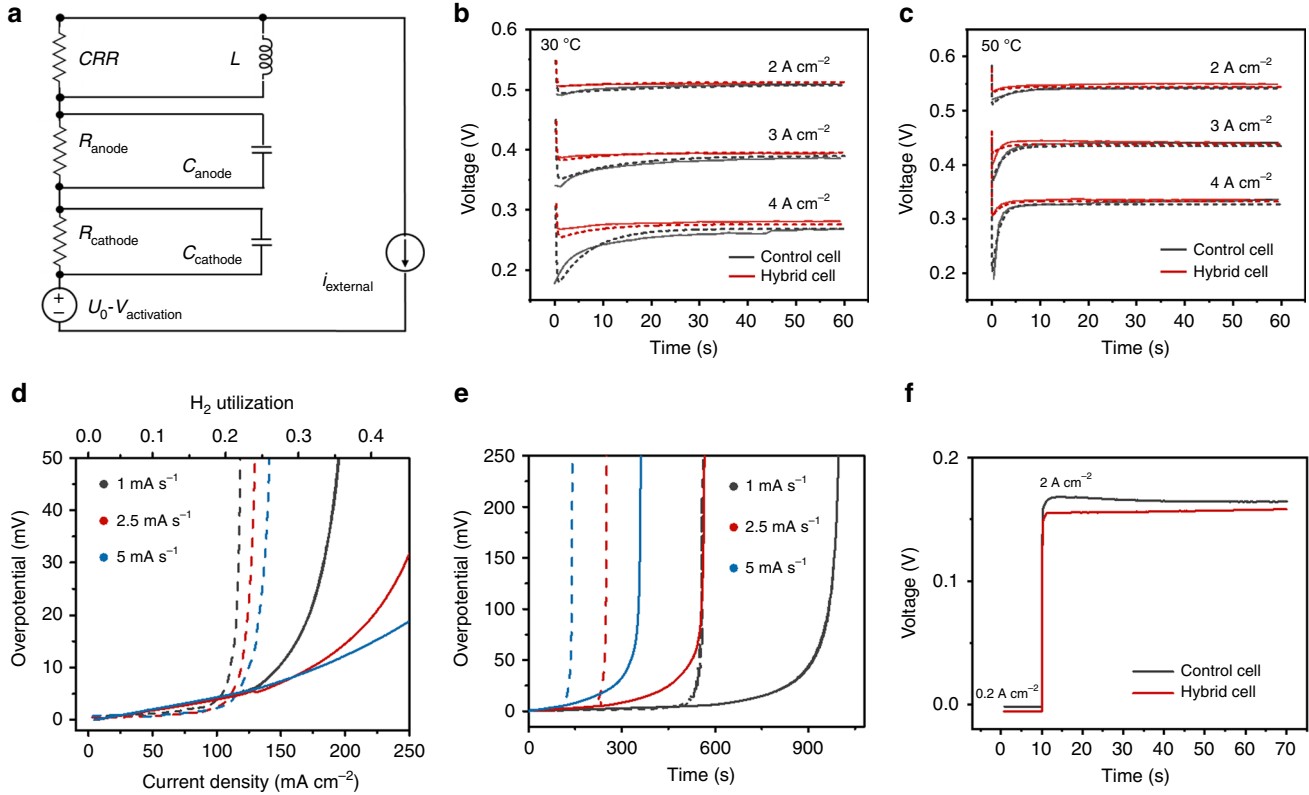

**Fig. 4 Origin of the enhanced transient performance attributed by the WO₃ layer. a** An equivalent circuit model for the control cell and hybrid cell. The values of CRR for the control cell (black) and hybrid cell (red) at time zero are listed in Supplementary Table 1. **b, c** Voltage–time profiles measured and fitted profiles of the hybrid cell and control cell upon switching the current density from 0.2 to 2 A cm$^{-2}$, 3, and 4 A cm$^{-2}$ at 30 °C (**b**) and 50 °C (**c**), respectively. **d, e** Anode overpotential of a hybrid cell (solid lines) and a control cell (dashed lines) vs. current density and H₂ utilization rate (**d**) and time (**e**) at different scan rates. 10 mL min$^{-1}$ H₂ (100% humidified) was fed to both the anodes and cathodes. **f** Voltage profiles of a hybrid cell and control cell in response to a current switching from 0.2 to 2 A cm$^{-2}$ at 30 °C. 50 mL min$^{-1}$ of H₂ (100% humidified) was fed to both the anodes and cathodes.

by a parallel unit of a resistor $R$ and a capacitor $C$, where $R$ is the resistor for the ohmic loss and $C$ is the equivalent capacitor due to the double-layer charging effect. A parallel connection of a current-responsive resistor (CRR) and an inductor $L$ is used to reflect the transient polarization that causes the power-output delay during transition operation[33]. The voltage–time profiles shown in Fig. 3c, d were then fitted using this model (see Methods for details). As shown in Fig. 4b, c, the transient profiles of the cells can be well fitted using this circuit model. The calculation also reveals that the hybrid cell shows two- to five-fold less CRR than the control cell (Supplementary Table 1). Upon switching to the current output of 4 A cm$^{-2}$ at 30 °C, the hybrid cell exhibits a near five-fold reduction of CRR, indicating that transient polarization has been reduced dramatically. Under a high current output ($I$), a significant reduction of the polarization ($R$) leads to a pronounced reduction of voltage drop ($R \cdot I$) and power loss ($R \cdot I^2$). In this context, the high $P$ and $E$ obtained can be attributed to the significant reduction of transient polarization upon the incorporation of the WO₃ layer.

To further probe the origin of the reduced polarization, both anode and cathode of the cells were connected to hydrogen and scanned with different current rates. The potential of the anodes (overpotentials) were obtained by subtracting the cell voltage with the ohmic loss as measured by an electrochemical impedance spectrum (EIS) technique. Figure 4d shows their overpotentials vs. current density, indicating that anode potential increases with current density linearly at low current range (e.g., <100 mA cm$^{-2}$). In the higher current range, overpotential rises exponentially with current density,

indicating the occurrence of fuel starvation. This observation is consistent with the Butler–Volmer equation. Compared with the control cell, the hybrid cell exhibits significantly higher overpotentials at low current densities, and the rise of overpotential is postponed significantly. Consistently, the hybrid cell shows significantly higher efficiency of hydrogen utilization, which is estimated by the current density and the flow rate of hydrogen. This finding suggests that the WO₃ layer could effectively regulate overpotential in response to the current-output demand, leading to more effective utilization of hydrogen and delayed occurrence of fuel starvation. Furthermore, for the hybrid cell, it takes 967, 557, and 353 s to reach an overpotential of 0.1 V under a scanning rate of 1.0, 2.5, and 5 mA s$^{-1}$, which is 1.7, 2.3, and 2.5 times longer than that of the control cell, respectively (Fig. 4e). This suggests that the WO₃ layer can also depress cell reversal, which may occur during high-power output and greatly deteriorate fuel cell performance.

The regulated overpotential in response to power demand affords the hybrid cell with significantly enhanced transient performance. Figure 4f shows both voltage profiles of the control and hybrid cell upon switching a current of 0.2 to 2 A cm$^{-2}$. The control cell exhibits higher voltage that is reduced with time, while the voltage profile for the hybrid cell is flat. In this configuration, the voltage measured is attributed from the cell resistance and the polarization developed during the current switching; whereas, the transition of the polarization is represented by the temporal evolution of the voltage. These observations are consistent with the power-output delay observed in Supplementary Fig. 8a.

Accordingly, the power-output delay observed in PEMFCs can be attributed to transient polarization, which is caused by the delayed proton-disassociation reaction and supply of protons to the cathodes. In response to a high-power demand, protons near cathodes are rapidly consumed, forming a concentration gradient that moves protons toward the cathodes and decreases the concentration of protons in anodes. The reduced proton concentration leads to an increase of anode potential, which promotes disassociation of hydrogen to provide more protons for the power demand. This power-regulating pathway relies on proton deficiency signaling via transport of protons through the cell. In this context, the power-output delay that commonly occurred in PEMFCs is caused by delayed response of the anodes to a power demand, which is intrinsically unavoidable for traditional PEMFCs. Nevertheless, the $WO_3$ layer could rapidly sense a power demand through the electric field between the anode and the cathode, and rapidly respond sequentially through releasing protons and electrons, increasing anode potential, and dynamically regulating the disassociation of hydrogen and supply of protons. In this sense, the $WO_3$ layer serves as a sensor for power demand and actuator for hydrogen disassociation, surpassing the traditional proton-transport based power-regulating pathway. Similar to a transistor, $WO_3$ analogically serves as the "gate", of which the gate voltage responds to the power demand, dynamically modulating the supply of protons in anodes ("source") to cathodes ("drain") to avoid the power-output delay.

**Fuel cells with enhanced power performance at low humidified conditions**. Sufficient humidification of inlet gases is critically important for PEMFCs[34]. Low humidification normally leads to increased internal resistance of the cells since proton conduction in the electrolyte membranes highly relies on their hydration condition[34,35]. In addition to degraded power output, uneven local water content in the electrolyte membranes would accelerate their mechanical degradation such as forming pinholes and cracks, which leads to cell failure[36,37]. However, high humidity condition complicates water management and increases the cost of a fuel cell system[38]. Thus, practical fuel cell devices are expected to operate at low humidified or even dry conditions. Although many efforts have been dedicated to develop electrolyte membranes with high water retention capability (e.g., incorporating $SiO_2$ and $TiO_2$ particles into Nafion films[39,40]), enhancing the performance and lifetime of PEMFCs at low humidified conditions with effectiveness and low cost remains challenging.

To further prove the viability of $WO_3$ in the hybrid cells, MEAs with a large size of $25\,cm^2$ were run under high-temperature and low humidity condition. Under 70 °C and a relative humidity of 50%, the control cell results in dramatically degraded performance, which is further deteriorated at the relative humidity of 30% (Supplementary Fig. 9). In contrast, the hybrid cell delivers significantly improved performance than the control cell, meanwhile the hybrid cell exhibits a similar performance at both humidity, indicating the adverse effects caused by low hydration can be effectively mitigated in the hybrid cells.

We also examined the cells under more realistic conditions (70 °C, relative humidity 30%, with air as the oxidant). Similar to above observation, the hybrid cell also delivers significantly improved performance at such harsh condition (Supplementary Fig. 9a). When switching the current outputs from 0.05 to $1\,A\,cm^{-2}$ and $1.5\,A\,cm^{-2}$, notable undershoots are observed for both the hybrid and the control cells due to low hydration of the ionomer in the membranes and in the $WO_3$ layer. Nevertheless, much less severe undershoots were observed in the hybrid cell (Supplementary Fig. 9b). Compared with the control cell, an extra average power

output of $307\,mW\,cm^{-2}$ was delivered by the hybrid cell in first 10 s when switching to the current output of $1\,A\,cm^{-2}$, which is ~70% of the steady-state power (Supplementary Fig. 9c). At the higher current output of $1.5\,A\,cm^{-2}$, the control cell fails to deliver the designed output power, whereas the hybrid cell still delivers reasonably high power (Supplementary Fig. 9b). Thus, hybrid cell could be operated in a more stable and reliable fashion during transient state compared with the control cell.

The significantly improved performance in the hybrid cell may be attributed to the $WO_3$ layer that dramatically improves the proton conductivity in dry condition. First, the presence of the $WO_3$/CNTs layer may help to retain water in the anode by slowing its evaporation upon being transported from the cathode. Second, $WO_3$ serves as both proton reservoir and conductor in the $WO_3$/CNTs/Nafion layer, facilitating the reactions at low humidity condition. More specifically, during an operation at dry condition (low relative humidity), the protons stored within the $WO_3$ (in the form of $H_xWO_3$) can be rapidly released, increasing the charge-carrier concentration. Meanwhile, $H_xWO_3$ are proton conductive, the presence of the $WO_3$ particles facilitates the transport of protons in the $WO_3$/CNTs/Nafion layer. Collectively, this could lead to increased proton conductivity in the hybrid cell, which help to initiate the cascade reactions ($H_2 + O_2 \rightarrow H_2O$) and produce water that subsequently hydrates the membrane and the $WO_3$ layer. The hydration of the hybrid cell, as a result, enables the operation of the cell even in dry condition. In this context, this design provides a simple yet highly effective approach to address the limitations associated with operations under low humidity condition, which has been a difficult challenge for the design and operation of fuel cell.

**Fuel cells with dramatically improved durability and reduced cost**. Beyond the improved transient performance, this design also dramatically improves the durability of PEMFCs against harsh operating conditions, such as fuel starvation, a main cause of degradation of PEMFCs. To demonstrate the improvement against fuel starvation, a hybrid cell and a control cell were operated under a constant current density of $0.2\,A\,cm^{-2}$, during which the feeding $H_2$ was switched to $N_2$ and cell voltage was recorded (Fig. 5a). For the control cell, the voltage drops rapidly below $-1.0\,V$ at ~2 s after the termination of hydrogen supply and continuously decreases with time. The observed cell-voltage reversal indicates that the anodic potential becomes more positive than the cathodic potential. Such a high anode voltage causes anode oxidation and catalyst aggregation (Supplementary Fig. 10), further deteriorating performance (Supplementary Fig. 11d). Consistently, as shown in Fig. 5c, the control fuel cell shows rapid peak-power decay, which is ~53% of the initial value after two rounds of fuel-starvation test (see Methods for testing conditions). Comparatively, the hybrid fuel cell shows much slower voltage decay; the occurrence of cell-voltage reversal is significantly delayed by ~6.5 s (Fig. 5a), indicating that the composite anode does improve durability against fuel starvation. Consistently, well-retained cell performance is observed for the hybrid cell after the fuel-starvation test (Fig. 5c). Fuel starvation also occurs during transient operations, such as the accelerating–deaccelerating process. To examine the improved durability against such transient operations, a control cell and the hybrid cell were subjected to oscillating current output between 50 and $1000\,mA\,cm^{-2}$ with a holding time of 120 and 30 s, respectively. The control cell exhibits a steady decrease in peak power by ~10% after 1000 testing cycles, in contrast to the unaltered performance of the hybrid cell, indicating improved durability against dynamic operating conditions (Fig. 5c), which is consistent with the depressed cell-voltage reversal shown in Fig. 4d.

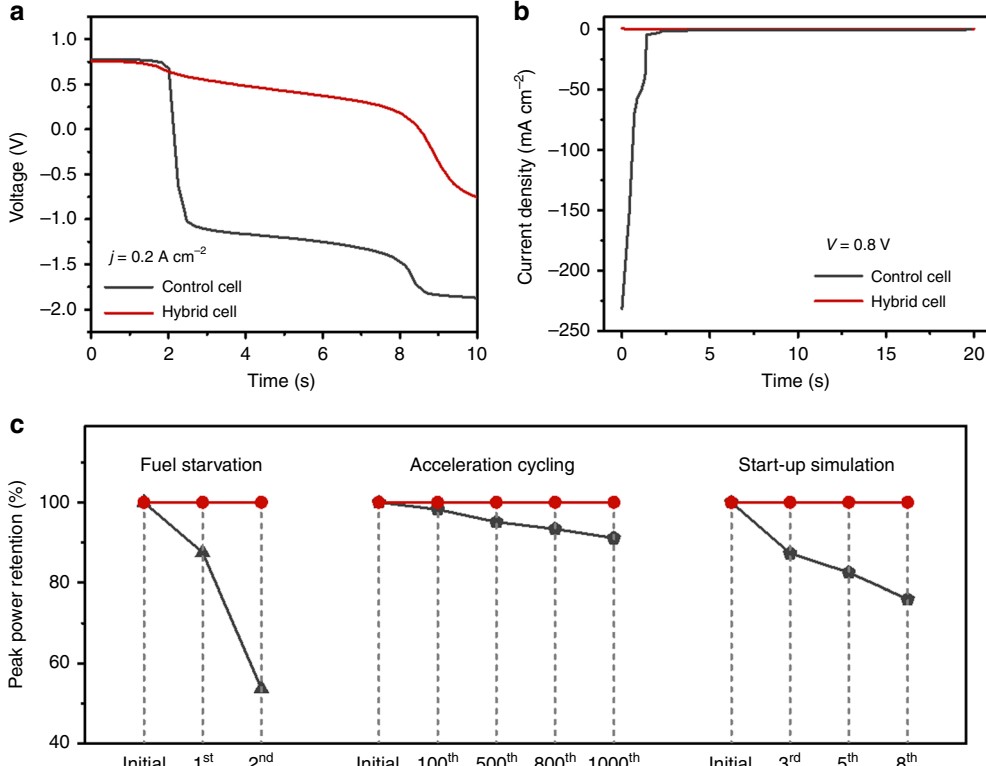

**Fig. 5 Hybrid fuel cells with improved durability against different operating conditions. a** Voltage profiles of a control cell and a hybrid cell operated under a constant current density of 0.2 A cm$^{-2}$ in response to switching the H$_2$ flow to N$_2$ flow. **b** Current profiles of a control cell and a hybrid cell operated under a constant voltage of 0.8 V in response to air intrusion to the anodes. **c** Retention of the peak-power density of three control cells and a hybrid cell after three accelerated stress tests (see the polarization curves in Supplementary Fig. 12). The hybrid cell was used throughout all three tests (red line). For the control cell testing (black line), one cell was used for each test. All of the tests were conducted at 50 °C, and detailed testing conditions are provided in the Methods.

Another noticeable cause of fuel cell degradation is the start-up process, during which residual air in anodes increases anode potential, which results in a dramatic increase of cathode potential if operated under a normal cell voltage. The increased cathode potential could lead to cathode oxidation that deteriorates cell performance and lifetime. The integrated rapid-response hydrogen reservoir (RRHR) can scavenge the oxygen effectively (reaction iv), stabilizing the anode and cathode potentials for the hybrid cell. To demonstrate this effect, a control cell and a hybrid cell were operated normally, during which the hydrogen flow was switched to nitrogen flow and 1 mL of air was injected to the cells, respectively. The control cell shows a gradual drop of OCV from 1.0 to ~0.44 V, while the hybrid cell still retains an OCV of 0.91 V after the air injection (Supplementary Fig. 12). It indicates that the intruding oxygen is thermodynamically favorable to react with H$_x$WO$_3$, which restraint the anode potential from increasing[29,30]. Figure 5b presents their current profiles upon being subjected to a constant voltage of 0.8 V. For the control cell, a large negative current over 230 mA cm$^{-2}$ is observed, indicating occurrence of cathode oxidation. Consistently, the peak power of such a control cell drops ~24% after eight cycles of start-up simulation test (Fig. 5c). For the hybrid cell, in sharp contrast, an initial discharging current of 0.35 mA cm$^{-2}$ is observed. The discharging current decreases with time; however, no cathode oxidation current could be observed, indicating improved durability against oxygen invasion to the anodes. Consistently, the hybrid cell shows negligible peak-power degradation after the start-up test (Fig. 5c).

Economically, as noted above, PEMFCs are often bundled with auxiliary energy-storage systems due to the intrinsic power-

output delay[8]. This design enables the fabrication of novel PEMFCs with dramatically enhanced power performance at low cost. This will reduce or even eliminate the use of expensive energy-storage systems and decrease costs at the system level. The improved power performance also enables the use of smaller fuel cells, but with required power for dynamic operations (e.g., forklifting). Taking the hybrid cell shown in Supplementary Fig. 8, an averaged $P$ of 152 mW cm$^{-2}$ for 10 s with a WO$_3$ loading of 5.1 mg cm$^{-2}$ could be provided, which corresponds to a specific power of 30 kW kg$^{-1}$ WO$_3$. Considering a fuel cell stack used for Toyota Mirai with an estimated anode area of ~18 m$^2$ (see Methods for details), a hybrid system with a similar WO$_3$ loading could be built using ~0.92 kg of WO$_3$. Currently, WO$_3$ is produced commercially at a cost of \$40/kg (the estimated cost is provided by Volta Materials, Inc., Culver City, CA, USA). With a negligible cost of <\$50, the hybrid PEMFCs could provide ~27.6 kW of extra power for 10 s, when the power output is raised from idle (below 10% peak power) to 100% peak power.

## Discussion

In summary, we have designed hybrid PEMFCs through integrating anodes with a layer of WO$_3$/CNTs composite, which serves as a rapid-response hydrogen reservoir, scavenger for oxygen, sensor for power demand, and regulator for hydrogen-disassociation reaction. Such multifunctional anodes afford PEMFCs with significantly improved transient power, performance under less hydrated condition, and durability against fuel starvation and transient operation. Such PEMFCs can be fabricated at a similar cost to that of traditional PEMFCs, yet provide a

significant cost reduction at the system level. Scale production of the PEMFCs can be readily achieved by using existing PEMFC-manufacture infrastructure and technologies. This work enables the production of PEMFCs with improved power performance, durability, lifetime, and cost for automotive and other applications.

## Methods

**Synthesis and characterization of WO₃/CNTs composites**. The composites of $WO_3$ and CNTs were synthesized through a hydrothermal process. A designed amount of $Na_2WO_4 \cdot 2H_2O$ (≥99% purity) and $(NH_4)_2SO_4$ (≥99.0% purity) were dissolved in 50 mL DI water, and 3 M $H_2SO_4$ was added to adjust the pH value to 1.5. CNTs were then dispersed in the solution by sonication to achieve a CNTs:WO₃ weight ratio of 1:4. The solution was then transferred to a 100 ml Teflon autoclave and reacted at 180 °C for 12 h. The resulting $WO_3$/CNTs composite was washed with DI water and dried for further use. Transmission electron microscopy (TEM) images were taken on a FEI Tecnai G2 F20 microscope operated at an accelerating voltage of 200 kV. X-ray diffraction (XRD) measurement was performed on a Rigaku X-2000 diffractometer using Cu Kα radiation with a Ni filter. The tube voltage and the tube current were maintained at 40 kV and 100 mA, respectively. The sample was scanned from 10° to 80° (2θ) at a scan rate of 5° min⁻¹. Thermogravimetric analysis (TGA, TA Instruments, USA) was conducted in an air environment. During the measurement, samples were initially held at 100 °C for 15 min to remove trapped water, and subsequently heated to 700 °C at a heating rate of 5 °C min⁻¹.

The electrochemical measurements of the composite were conducted on a BioLogic VMP-300 multipotentiostat workstation (Bio-Logic Science Instruments, France). The electrolyte solution was 0.5 M $H_2SO_4$. Ag/AgCl and platinum foil were used as the reference electrode and the counter electrode, respectively. To fabricate the electrodes, WO₃/CNTs composite and PVdF (Kynar) binder were mixed in a mass ratio of 9:1 and homogenized in N-methyl-2-pyrrolidone (NMP) to form slurry. The homogenous slurry was coated on Ti foil substrates and dried at 90 °C for 6 h under vacuum. As-formed electrodes were then pressed under a pressure of 2 MPa.

**Evaluation of energy-storage ability of WO₃/CNTs in MEA**. WO₃/CNTs composite electrodes were fabricated using the following procedure. The composite was mixed with Nafion solution (5 wt%, Dupont, USA) at a 9:1 weight ratio to form slurry, which was sprayed on a PTFE-treated carbon paper (Toray TGH-060) to prepare the composite electrodes with desired WO₃ loadings. Cells were assembled using Aquivion® R79-02S as the membrane electrolytes. Pt/C cathodes were prepared by spraying a homogeneous ink, which was prepared from commercial Pt/C (40 wt% Pt, JM) and Nafion solution (5 wt%, Dupont) in ethanol on the membrane to achieve a Pt loading of 0.4 mg cm⁻². The MEAs were then made by sandwiching the membrane electrolyte between a WO₃ composite electrode and a gas diffusion layer (GDL, prepared by spraying XC-72 carbon black on Toray carbon paper with a loading of 2 mg cm⁻²). The structure of MEA is also schematically shown in Supplementary Fig. 3.

Electrochemical measurements were carried out on a Solartron 1860/1287 Electrochemical Interface (Solartron Analytical). The charging process was conducted by supplying the Pt/C cathodes with hydrogen (0.1 L min⁻¹) and the WO₃ anode with nitrogen (0.1 L min⁻¹). The cells were charged to 0 V (vs. DHE) using a constant current of 0.2 A g⁻¹. The inlet gas of cathode was then switched to oxygen (0.1 L min⁻¹) (see Supplementary Fig. 3). The cells were discharged at different rates.

**Fabrication and testing of fuel cells**. Catalyst ink was prepared by ultrasonically blending a commercial Pt/C catalyst (40 wt% Pt, JM) with a Nafion solution (5 wt%, Dupont) and ethanol for 2 h, which was sprayed onto both sides of the proton-exchange membranes (Aquivion® R79-02S) with a Pt loading of 0.05 and 0.4 mg cm⁻² in the anode and cathode, respectively. Hybrid MEAs were fabricated by sandwiching the catalyst-coated membranes between a WO₃ composite electrode and a pre-fabricated GDL, and then hot-pressed together at 130 °C under 0.1 MPa for 1 min. Control MEAs were fabricated using a GDL instead of the WO₃ electrode at the anode side. The cells were tested under different conditions listed below:

1. *Steady-state tests*: steady-state tests were conducted in a 5 cm² single cell at 30 and 50 °C using fuel cell test stations (Fuel Cell Technologies, Inc.). The anodes were fed with $H_2$ at a stoichiometry of 1.5. The cathode was supplied with $O_2$ at a stoichiometry of 4. The fuel (hydrogen) and the oxidant (oxygen) were humidified by bubbling through distilled water to achieve 100% humidity. The cell resistance was measured by a current interrupt technique during steady-state and dynamic response tests.

2. *Dynamic response tests*: while the cells were running under a steady state, the current outputs of the cells were abruptly increased from 0.05 to 2, 3, and 4 A cm⁻². The test was conducted by holding under each of the target current densities for 60 s. The anodes were fed with $H_2$ at a stoichiometry of 1.5. The cathode was supplied with $O_2$ at a stoichiometry of 4. The fuel

(hydrogen) and the oxidant (oxygen) were humidified by bubbling through distilled water to achieve 100% humidity.

3. *Simulated durability tests*: three control cells were used (one for the starvation test, one for the acceleration/deceleration test, and one for the start-up test), while only one hybrid cell was used for the three tests. All tests were conducted at 50 °C.

   a. The simulated $H_2$ starvation tests were performed by switching the $H_2$ supply to $N_2$ flow (0.2 L min⁻¹) to a control cell and a hybrid cell while operated under a constant current density of 0.2 A cm⁻² using the external direct-current power source (1287A Potentiostat Galvanostat). The cell voltage was recorded during the measurement, and each cycle lasted for 10 s after switching to $N_2$ flow. I–V curves were measured after each cycle, and the peak-power density was recorded.

   b. The simulated acceleration and deceleration tests were conducted by oscillating the current output of a control cell and the hybrid cell between 50 and 1000 mA cm⁻² with a holding time of 120 and 30 s, respectively. I–V curves were measured after certain cycles, and the peak-power density was recorded.

   c. For start-up tests, a control cell and the hybrid cell were kept at open circuit while fed with $H_2$ and $O_2$. The gas for the anodes was switched from $H_2$ to $N_2$ (30 mL min⁻¹) for 10 s to purge the anode compartment. Then, 1 mL of air was injected into the anodes, and the equilibrium OCV of the cells was recorded by the Solartron 1287 electrochemical interface. The cells were then maintained at a constant voltage of 0.8 V for a period of 20 s, and their current profiles were recorded. The simulated start-up process was conducted by maintaining a control cell and the hybrid cell at a constant voltage of 0.8 V. $H_2$ supply was then switched to an air flow of 20 mL min⁻¹. I–V curves after each two cycles were measured, and the peak-power density was recorded.

**Understanding the origin of the transient behaviors**. To avoid the influence of cathodes during transient operation, symmetric cells were used for testing. The MEAs were prepared according to the same method as other cells. The Pt loadings on the anode and cathode were both 0.4 mg cm⁻². The hybrid cell was fabricated with a WO₃ composite electrode (WO₃ loading of ~4.9 mg cm⁻²) as the anode, while the control cell was fabricated with a prefabricated GDL. During the scan current tests, 10 mL min⁻¹ $H_2$ (100% humidified) was fed into the anodes and cathodes. The voltage of the cells was recorded by a Solartron 1860 Electrochemical Interface (Solartron Analytical) under scan rates of 1, 2.5, and 5 mA s⁻¹, respectively. The cut-off voltage of the test was set to 0.3 V to avoid oxidation of the anodes. For the transient-current tests, the cell voltage was monitored by a Solartron 1860 electrochemical interface, when the output current was switched from 0.2 to 2 A cm⁻², while 50 mL min⁻¹ of $H_2$ (100% humidified) was fed into both the anodes and cathodes. The cell resistance was measured by impedance spectroscopy (Solartron 1860/1287 Electrochemical Interface, Solartron Analytical) at open circuit; an AC signal of 10 mA was applied to the cell, and the frequency range of the impedance tests was limited from 10⁵ Hz down to 1 Hz.

**Fitting of the voltage transient profiles with the equivalent circuit model**. The fitting of the transition profiles was conducted using MATLAB based on the equivalent circuit model proposed. Based on the voltage profiles of the cells upon switching from a low current density to a high current density, cell voltage at steady state ($U_{stable}$) was obtained. Based on $U_{stable} \approx U_0 - V_{activation} - I \cdot R_{eo}$, where $U_0$ is the thermodynamic voltage of the cell, $V_{activation}$ is the activation loss, $R_{eo}$ is the overall equivalent *ohmic* resistance ($R_{anode} + R_{cathode}$), and $I$ is the current (A/cm²), $U_0 - V_{activation}$ and $R_{eo}$ were obtained.

Based on $U_{undershoot} = \Delta I \cdot CRR$, where $U_{undershoot}$ is the voltage undershoot measured and $\Delta I$ is the step of current change from 0.05 to 2 A cm⁻², 3 and 4 A cm⁻², respectively, a series of CRR were obtained. Assuming that CRR is a function of $I$ and temperature $T$, following a relation of $CRR = a + b \cdot I + c \cdot I^2 + d \cdot T$, where $a$, $b$, $c$, and $d$ are constants, CRR vs. $I$ and $T$ were obtained using a method of least squares. Combinations of $R_{anode}$, $R_{cathode}$, $C_{anode}$ and $C_{cathode}$, and inductance ($L$) were then used to fit voltage profiles based the circuit model. Related parameters and fitted parameters are provided in Supplementary Table 2. Note that the double-layer charge effect reflected by $R_{anode} \cdot C_{anode}$ and $R_{cathode} \cdot C_{cathode}$ generally exhibit a delay within a time constant of 1 s[33]. Consistently, it was found that variations of $R_{anode}$, $R_{cathode}$, $C_{anode}$, and $C_{cathode}$ do not affect the fitted profiles significantly. As-observed delay during the transition operation of fuel cells is mainly attributed from the CRR and $L$.

**Estimation of the total cell area of a Mirai fuel cell stack**. Based on the information released by Toyota, the fuel cell stack in Mirai is 37 L in total volume (including fastener), and consists of 370 cells (1.34 mm in thickness) in single-line stacking. Accordingly, the maximal cell area afforded by such a fuel cell stack would be 37 L/(370 × 1.34 mm) = 746 cm². Considering the volume of ancillary components and our private communications with a fuel cell manufacturer, we estimate the cell area to be ~500 cm² and the total anode area to be 500 cm² ×

$370 = 18.5\ m^2$. Thus, we use an estimated total area of $18\ m^2$ to assess the impact of incorporating the $WO_3$ layers.

## Data availability

The data that support the findings of this study are available from the corresponding author upon request. The source data underlying Fig. 2e–h, 3a–f, 4b–f, and 5a–c and Supplementary Figs. 1, 2, 4–9, 11, and 12 are provided as a Source Data file.

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

## Acknowledgements

We are grateful to the funding support from the Department of Energy (DOE) ARPA-E program and the UCLA ENN Center for Nanomedicine and Energy Conversion.

## Author contributions

G.Shen, J.Liu, and F.L. synthesized and characterized the materials. G.Shen and H.B.W. designed and carried out the electrochemical measurement. G.Shen and J.Liu fabricated MEAs and conducted fuel cell tests. P.X. built up equivalent circuit and performed calculations. C.T. and K.J. helped with high-temperature fuel cell measurement. J.Li, M.L., M.C., J.P.L., G.Soloveichik, and H.L. contributed to the discussion and analysis of the experimental results. G.S., H.B.W., and Y.L. wrote the paper. J.Z. and Y.L. conceived the idea and supervised the project.

## Competing interests

The authors declare no competing interests.
