## [Peer Review File · Nature Communications]

Reviewers' comments:

Reviewer #1 (Remarks to the Author):

The authors describe an original way to improve the stability of PEMFC during start/stop experiments and fuel starvation by integrating a layer of WO₃ between the gas diffusion layer and the catalyst layer, at the anode side. The authors well described the synthesis, nanostructural and electrochemical characterizations of the WO₃ with Carbon Nanotubes. They also give the details of the integration of this compound in the membrane electrode assembly. According to the authors, this layer allows to make the PEMFC more resilient to transient power demand. The extensive electrochemical characterizations clearly and smartly support their conclusions. Considering the details in the experimental part, we can be confident that this works gives fruitful information and sufficient to reproduce the fabrication, the integration and the evaluation in real operation of this WO₃ based layer. The approach is novel and of interest for the PEMFC community and also to a broader audience for researcher working of electrochemical energy conversion devices with proton conducting electrolyte. Thus, I would strongly recommend to publish this article, after some minor revisions.

In Figure 1, the authors give the electrochemical reaction of WO₃ in acidic conditions (equation (iii)). Could the author give, somewhere in the manuscript or in the supplementary information, the standard redox potential of this reaction and relate it to the Oxidation/Reduction peaks on the cyclic voltammograms shown in Figure 2F.

The fast response of the WO₃/CNT layer was evaluated in a PEMFC like configuration but without anode catalyst layer. The WO₃/CNT is mixed with Nafion to make a porous composite layer on the anode Gas Diffusion Layer (GDL) and it is in direct contact with the membrane (details given in the supplementary information). In this case, the WO₃ is in the presence of acidic water when hydrated thanks to the proton coming from Nafion. This is not absolutely clear in the description, but the same composition of the layer (WO₃/CNT layer+Nafion) seems to be used in real PEMFC. Could the author state it more clearly?

The whole mechanisms rely on the possibility to WO₃ to exchange protons with the membrane. This explains why the authors mixed this compound with ionomer. When water activity decreases the proton resistance of this layer decreases and this makes the whole electrochemical reaction less facile. This could make this layer less efficient in drier conditions. Could the authors comment? Experiments in fuel cell have been conducted in fully humidified conditions, the impact of the results (already important) would have been greatly enhanced if additional experiments in drier conditions, e.g. lower RH along with higher temperature (>50°C), had been performed. If the authors have conducted such experiments, could they provide the results? Is this lower hydration of the ionomer in the WO₃/CNT layer could explain the decrease in efficiency of the strategy when the temperature is raised from 30 to 50°C? Could the authors comment?

The authors wrote: "This design enables the fabrication of novel PEMFCs with dramatically enhanced performance at low cost". This statement seems to me a bit strong. Indeed, the performance demonstrates with the polarization curve on Figure S8 is reasonably good but not really at the state-of-the-art. The layer does not allow to increase steady state performance as shown on Figure 3C. I agree that the stability upon start-up/shut-down and fuel starvation is enhanced. The resilience to transient is also improved thanks to this layer but it does not allow to increase the steady-state performance (of course, this is not the primarily objective). This is true that the initial objective for which this layer has been designed is reached. However, experiments with Air at lower relative humidity (e.g. 70%RH or lower) and at higher temperature (e.g. 70°C) would have definitely demonstrated the interest of this layer. It seems to me that this observation of the authors should be nuanced. Note, that it does not call into question the originality, the relevance and the quality of their work.

Could the authors explain why they did use CNT instead of carbon black? Is this possible to integrate WO₃ in the PEMFC with carbon black which is more conventionally used in PEMFC than CNT? I maybe missed something but it does not seem that the cost of CNT has been integrated to the calculation of the cost of the composite layer made of WO₃. Could the authors comment?

A short bibliography study on the use of WO₃ in PEMFC as well as on the mitigations strategy to

reduce the impact of fuel starvation and start-up/shut-down would have been useful in the introduction.

In figure 3, E and F seems to me a bit redundant, as well as H and I.

The authors may refer to a patent number in case their strategy has been patented.

Arnaud Morin
CEA, Grenoble

Reviewer #2 (Remarks to the Author):

The manuscript presents a thorough electrodynamic assessment of the transient performance of WO₃-modified PEMFC anodes to address continuity of power delivery following practical operational issues such as variations in power demand, fuel starvation and start-up. The rationale for the study is based on the variable availability of protons under transient polarisation of conventional Pt/C anodes with the proposed solution being the use of an integrated hexagonal tungsten bronze layer as a proton reservoir.

Several studies have reported on the properties of WO₃/H_xWO₃ as catalysts relevant for fuel cell applications, particularly the well-known potential dependent intercalation/de-intercalation of H⁺ ions and the increased electronic conductivity of the tungsten bronze (H_xWO₃) [e.g. Liu, et al, Int J Hydrogen Energy, 43, (2018), 8944; Micoud, F., et al, Phys. Chem. Chem. Phys., 12, (2010), 1182; Wickman, B., et al, Electrochimica Acta 56 (2011) 9496, in addition to refs 21 and 22 of the current manuscript]. These properties are confirmed by the CV (Fig 2B), the H⁺ supply /consumption by the tungsten bronze being driven by the polarisation of the bronze (anode) under the different fuel cell conditions described. Charge balance in the bronze is maintained by the oxidation/reduction of tungsten ions (W⁵⁺/W⁶⁺) and the accompanying electronic charge transfer between the bronze and the anode current collector. Consequently, variation of the anode potential, induced by operational conditions, will influence the direction and magnitude of the reaction described in Fig 1 reaction (iii) which accounts for much of the reported cell behaviour.

A H_xWO₃ supported ORR reaction is suggested by Fig 1 reaction (iv) but its viability, or details of any required conditions, e.g. potential, is not discussed or supported by any mechanism or cited literature. Hernández-Pichardo, et al, [Int J Hydrogen Energy, 40, (2015), 17371] proposes an ORR mechanism requiring the presence of Pt (as is available in the current study at the interface between the WO₃ layer and the PtC electrode); note that this requires anode polarisation to (0.8V vs NHE).

The nature of the WO₃-Pt interaction therefore becomes of interest. The proximity of Pt and WO₃ was shown to be important in promoting the beneficial role of WO₃ in the CO tolerance of Pt [Micoud, F., et al, Phys. Chem. Chem. Phys., 2010, 12, 1182; Micoud, F., et al, Electrochem. Commun., 2009, 11, 651–654]. This aspect of the performance dependence is not addressed but may be significant.

The subsequent measurements support the general concepts presented. Although WO₃-modified electrodes exhibit similar polarisation curves to the control system (Fig 3C), they are more responsive under current switching and the analyses through modelling supports the reduction of transient polarisation which could be accounted for by the enhanced availability of protons from the WO₃ layer.

The response of fuel cells to power demand is increasingly important in their development for automotive applications and this manuscript directly addresses this through a practical solution. Whilst further discussion might be expected on mechanisms and constraints, the general principles illustrated could have an important impact in the field.

Response to Reviewers**Response to Reviewer 1:**

Q1: In Figure 1, the authors give the electrochemical reaction of WO_3 in acidic conditions (equation (iii)). Could the author give, somewhere in the manuscript or in the supplementary information, the standard redox potential of this reaction and relate it to the Oxidation/Reduction peaks on the cyclic voltammograms shown in Figure 2F.

A1: We highly appreciate the reviewer's suggestion. In the revised manuscript, the standard redox potential for the equation (iii) was provided (Figure 1). According to the paper by Michael Hitchman (Analysis of equilibrium potentials of hydrogen tungsten bronzes, *J. Electroanal. Chem.*, 85 (1977), 135-144), the standard potentials of the related reactions are listed below:

The two pairs of redox peaks observed (the peaks at 0.23 V and 0.34 V, and the peaks at -0.18 V and -0.02 V) in Fig. 2F account for reversible insertion of protons in the channels and at the surface sites of WO_3 crystals, respectively (*Nano Letters*, 15, 4834-4838 (2015)).

Meanwhile, the oxidation of HWO_3 by oxygen is a thermodynamic favored process with an equilibrium constant (K) of 2.2×10^{31} (*J. Electroanal. Chem.*, 85 (1977), 135-144). HWO_3 exhibits blue color, which has been chromically used for hydrogen sensing. Exposing blue HWO_3 to air (O_2) allows a spontaneous oxidation reaction (Reaction iv), which bleaches the blue color. In contrast, keeping a blue HWO_3 sample in vacuum show no change in color over several months. The related information can also be found in the report by Hajime Takahashi *et al* (Optimization of hydrogen sensing performance of Pt/ WO_3 gasochromic film fabricated by sol-gel method, *Sensors and Materials*, Vol 29, 9 (2017), 1259-1268).

Q2: The fast response of the WO_3/CNT layer was evaluated in a PEMFC like configuration but without anode catalyst layer. The WO_3/CNT is mixed with Nafion to make a porous composite layer on the anode Gas Diffusion Layer (GDL) and it is in direct contact with the membrane

(details given in the supplementary information). In this case, the WO_3 is in the presence of acidic water when hydrated thanks to the proton coming from Nafion. This is not absolutely clear in the description, but the same composition of the layer (WO_3/CNT layer+Nafion) seems to be used in real PEMFC. Could the author state it more clearly?

A2: The fast response of the WO_3/CNT layer was evaluated in a PEMFC-like configurations but without an anode catalyst layer. This experiment aims to demonstrate the proton-storage capability of WO_3 in MEA configuration, during which protons generated from the cathode were transported to the anode through the Nafion in the membrane and the $\text{WO}_3/\text{CNT}/\text{Nafion}$ layer. The same $\text{WO}_3/\text{CNT}/\text{Nafion}$ layer was used as the GDL in the real PEMFC, where the mechanism has been described in the manuscript. We revised the statement in the revised manuscript. Thank you for the suggestion.

Q3: The whole mechanisms rely on the possibility to WO_3 to exchange protons with the membrane. This explains why the authors mixed this compound with ionomer. When water activity decreases, the proton resistance of this layer decreases, and this makes the whole electrochemical reaction less facile. This could make this layer less efficient in drier conditions. Could the authors comment?

A3: Thank you for this highly intuitive comment. Generally, the transport of protons in Nafions is mainly through the water channels within the membranes, whereas dry conditions generally reduce proton conductivity. In sharp contrast, the transport of protons in hexagonal $\text{WO}_3 \cdot (\text{H}_2\text{O})_{0.5}$ is mainly through the single-file water chains within the crystals, whereas the rate of insertion/extraction of protons is much less affected by the water activity (Hierarchical Nanostructured WO_3 with Biomimetic Proton Channels and Mixed Ionic-Electronic Conductivity for Electrochemical Energy Storage. *Nano Letters* **15**, 6802-6808 (2015)). Adding ionomer to the WO_3/CNT layers is to provide efficient proton pathways among the WO_3 particles. Since the conductivity of protons in the ionomer is affected significantly by the water activity, decreasing water activity could decrease the conductivity of the WO_3/CNT layer.

Nevertheless, the WO_3 crystals within the WO_3/CNT layer could serve as both rapid-response reservoir and conductor of protons, which could increase proton concentration (charge-carrier concentration) and facilitate their transport, particularly during dry conductions. In this context, the WO_3/CNT layer can significantly mitigate the effects associated with operations in dry condition. As shown in Fig R1 and R2, the incorporation of WO_3/CNT layer dramatically improves the power output of hybrid cells operated at high temperature (70°C) and less humid condition (30% and 70% relative humidity). Furthermore, the structure of $\text{WO}_3/\text{CNT}/\text{Ionomer}$ could be engineered and optimized, which could further mitigate the effects associated with operation in dry conditions.

Q4: Experiments in fuel cell have been conducted in fully humidified conditions, the impact of the results (already important) would have been greatly enhanced if additional experiments in drier conditions, e.g. lower RH along with higher temperature ($>50^\circ\text{C}$), had been performed. If

the authors have conducted such experiments, could they provide the results? Is this lower hydration of the ionomer in the WO₃/CNT layer could explain the decrease in efficiency of the strategy when the temperature is raised from 30 to 50°C? Could the authors comment?

A4: We have conducted experiments at 70°C using a set of larger MEAs (25 cm²) under a relative humidity (RH) of 30% and 50%, respectively. Fig. R1 (Supplementary Fig. 9 in the revised manuscript) shows the polarization curves of a hybrid cell and a control cell, indicating the significantly improved performance of the hybrid cell at the high-current-density region. The hybrid cell gives almost identical performance under the both relative humidity, whereas the performance of the control cell at 30% RH is significantly worse than that at 50% RH.

Fig. R1 Polarization curves of a hybrid cell (solid lines) and a control cell (dashed lines) at 70 °C under relative humidity of 30% and 50%, respectively. Humidified H₂ (stoichiometry = 1.5) and O₂ (stoichiometry = 4) were fed to the anodes and cathodes.

To further quantify the effect of WO₃/CNT layer on the transient performance, we also examined the response of the cells to a constant current output (2 and 3 A cm⁻²). Fig. R2 shows the corresponding voltage and the difference of the power output (ΔP) of the hybrid cell and the control cell. Notable undershoots are observed in the both cells, which is expected due to low hydration of the ionomer in the membranes and in the WO₃ layer. Nevertheless, during the transition period (0 - 5 s), the power provided by the hybrid cell is significantly higher than that of the control cell (Fig. R2C and D). Especially in the drier condition (30% RH), compared with the control cell, the hybrid cell can provide extra instantaneous power of 400 mW cm⁻² when the current is switching to 3 A cm⁻². After reaching steady state, the hybrid cell still

provides significantly higher power output than the control cell ($\sim 350 \text{ mW cm}^{-2}$), which is near 1/3 of the maximum power of the control cell. This study suggests significantly improved performance can also be achieved at less high temperature and hydrated state, which is particularly important for practical applications.

Fig. R2 Voltages and ΔP of the control cell and the hybrid cell upon switching the current output from 0.05 A cm^{-2} to different current outputs at $70 \text{ }^\circ\text{C}$ under relative humidity of 50% (A, C) and 30% (B, D).

The significantly improved performance in the hybrid cell may be attributed to the WO_3 layer that dramatically improves the proton conductivity in dry condition. Firstly, the presence of the WO_3/CNT layer may help to maintain the humidity in the anode by slowing the evaporation of water transported from the cathode. Secondly, WO_3 serves as both proton reservoir and conductor in the $\text{WO}_3/\text{CNT}/\text{Nafion}$ layer, which may initiate the reaction even at low humidity condition.

More specifically, during operation at dry condition (low relative humidity), the protons stored within the WO_3 (in the form of H_xWO_3) can be rapidly released, increasing the charge-carrier concentration. Meanwhile, WO_3 particles are proton conductive, shortening the diffusion length of protons in the $\text{WO}_3/\text{CNT}/\text{Nafion}$ layer. Collectively, the increased proton conductivity in the hybrid cell initiates the cascade reactions ($\text{H}_2 + \text{O}_2 \rightarrow \text{H}_2\text{O}$), producing more

water to hydrate the membrane and the WO_3 layer, which leads to significantly improved performance in dry condition. This design provides a simple yet highly effective approach to address the low humidity limitation, which has been an engineering challenge for fuel cell design and operation.

The decrease in efficiency of the strategy when the temperature is raised from 30 to 50°C could also be associated with proton transport, humidity control and other engineering factors, which may require more systematic studies.

Q5: Experiments with Air at lower relative humidity (e.g. 70%RH or lower) and at higher temperature (e.g. 70°C) would have definitely demonstrated the interest of this layer. It seems to me that this observation of the authors should be nuanced.

A5: We conducted experiments with air at 70°C and 30% relative humidity. The results were presented in Supplementary Figure 9. Under such harsh and more realistic conditions, significant voltage undershoots or even failure to deliver designed power output was observed in control cell. In contrary, even though voltage undershoots are not fully avoided due to insufficient hydration of ionomers in WO_3 layer as discussed earlier, the power output during transient period and steady state is much improved in hybrid cell.

Q6: Could the authors explain why they did use CNT instead of carbon black? Is this possible to integrate WO_3 in the PEMFC with carbon black which is more conventionally used in PEMFC than CNT? I maybe missed something but it does not seem that the cost of CNT has been integrated to the calculation of the cost of the composite layer made of WO_3 . Could the authors comment?

A6: CNTs were used in this work to take advantages of the high electronic conductivity and high chemical stability, in comparison with the conventional conductive carbon. Meanwhile, CNTs generally have a long aspect ratio, which allow them to form composites with the WO_3 nanorods with intimate interface, allowing more effective charge transport.

Please note that the price of CNT has been substantially reduced in recent years (~ \$15-40/kg in 2019), which is lower than the cost of WO_3 (~ \$40/kg). Although the cost of CNT was not included for the original estimation of the cost of composite layer, considering the content of CNTs in the composite layer (20 wt-%), it is estimated that the total material cost for the WO_3 /CNT layers for a Toyota Mirai fuel cell stack will be ~ \$50, which is quite low compared with the costs of other materials used.

Q7: A short bibliography study on the use of WO_3 in PEMFC as well as on the mitigations strategy to reduce the impact of fuel starvation and start-up/shut-down would have been useful in the introduction.

A7: WO_3 has been extensively explored as co-catalysts or catalyst supports for fuel cells, which could promote HOR reactions through hydrogen spillover effect and mitigate the CO-poisoning

of Pt catalysts. The conventional strategies to solve the problems of fuel starvation and start-up/shut-down include the use of more corrosion-resistant non-carbon supports, the deployment of a water oxidation electrocatalyst in the anode, and other strategies. This study appears to be the first report of using WO_3 to improve the dynamic performance of fuel cells. In the revised manuscript, we have added a short review on the use of WO_3 in PEMFC as well as on the mitigation strategy to reduce the impact of fuel starvation and start-up/shut-down in the introduction section.

Q8: In figure 3, E and F seems to me a bit redundant, as well as H and I. The authors may refer to a patent number in case their strategy has been patented.

A8: We thank the reviewer for the valuable suggestion. Figure 3 have been revised accordingly. Our related patent was cited on page 2 as ref. 24.

Response to Reviewer 2:

Q1: A H_xWO_3 supported ORR reaction is suggested by Fig 1 reaction (iv) but its viability, or details of any required conditions, e.g. potential, is not discussed or supported by any mechanism or cited literature. Hernández-Pichardo, et al, [Int J Hydrogen Energy, 40, (2015), 17371] proposes an ORR mechanism requiring the presence of Pt (as is available in the current study at the interface between the WO_3 layer and the PtC electrode); note that this requires anode polarisation to (0.8V vs NHE). $H_xWO_3 = xH^+ + WO_3 + xe^- + Pt-O_2 = Pt-O_2H$

A1: Thank you for this wonderful question. According to Michael Hitchman (Analysis of equilibrium potentials of hydrogen tungsten bronzes, *J. electroanal. Chem.*, 85 (1977), 135-144), the reaction $2HWO_3 + 1/2O_2 = 2WO_3 + H_2O$ exhibits a standard redox potential of 0.958 V ($E^0 = 0.958$ V vs a standard hydrogen electrode). The oxidation of HWO_3 by oxygen is a thermodynamic favored process with an equilibrium constant (K) of 2.2×10^{31} .

The insertion of protons to WO_3 results in the formation of blue color HWO_3 . Such a chromic transition is commonly used for hydrogen sensing. Exposing HWO_3 to air (O_2) allows the spontaneous oxidation of HWO_3 , which bleaches the blue color. In contrast, keeping a blue HWO_3 sample in vacuum show no change in color over several months. The related information can also be found in the report by Hajime Takahashi et al (Optimization of hydrogen sensing performance of Pt/ WO_3 gasochromic film fabricated by sol-gel method, *Sensors and Materials*, Vol 29, 9 (2017), 1259-1268).

Hernandez-Pichardo reported the synthesis of WO_3 on Pt/C, which required an anode polarization of 0.8V vs NHE. In this work, XRD and TEM studies suggest the absence of WO_3 crystalline structure in their catalysts; they could not observe any WO_3 particles by TEM either, although EDX indicates the presence of W element. Based on the observation, the authors concluded that as-formed WO_3 was present as small clusters of mono and polytungstates, which are totally different in comparison with our nanocrystalline WO_3 with well-defined proton conductive channels. Considering the dramatic difference in composition and structure of tungsten oxide, it is not surprised that the WO_3 reported by Hernandez-Pichardo exhibited dramatically different electrochemical behaviors, in comparison with the nanocrystalline WO_3 used herein.

Q2: The nature of the WO_3 -Pt interaction therefore becomes of interest. The proximity of Pt and WO_3 was shown to be important in promoting the beneficial role of WO_3 in the CO tolerance of Pt [Micoud, F., et al, *Phys. Chem. Chem. Phys.*, 2010, 12, 1182; Micoud, F., et al, *Electrochem. Commun.*, 2009, 11, 651–654]. This aspect of the performance dependence is not addressed but may be significant.

A2: We appreciate the reviewer's suggestion. The interactions between WO_3 and Pt has been extensively studied previously. WO_3 has also been used as a co-catalysts or catalyst supports for fuel cells, which could promote the HOR and improve the tolerance of Pt for CO. The focus

of this work is the integration of WO_3 in the anodes, which serves as a rapid-response hydrogen reservoir, oxygen scavenger, sensor for power demand, and regulator for hydrogen-disassociation reaction. In such a design, the WO_3 is present in the WO_3/CNT layer next to the Pt/C catalytic layer, rather than as co-catalyst nor as catalyst support for Pt. In our future study, we will disperse Pt within the WO_3/CNT layer and investigate their CO tolerance.

Q3: The response of fuel cells to power demand is increasingly important in their development for automotive applications and this manuscript directly addresses this through a practical solution. Whilst further discussion might be expected on mechanisms and constraints, the general principles illustrated could have an important impact in the field.

A3: Thank you for your kind comment and encourage. Indeed, a fuel cell is highly complicated system. Similarly, dynamic response of fuel cells to power demand is also quite a complicated problem, which is associated with multiple factors such as gas transport, water management, and other engineering factors. The main goal of this work is to present a potential solution for the dynamic-response limitation, mainly from the aspect of material design and synthesis. We also added more discussion regarding the mechanism and constraints of this strategy to the revised manuscript. We recognize that much more need to be done in order to understand the mechanism and constraints, which will be conducted systematically in our future research. We hope that this work can inspire more colleagues in the community to work together and tackle this challenge. Thank you!

REVIEWERS' COMMENTS:

Reviewer #1 (Remarks to the Author):

The manuscript has been improved according to previous comments and I would like to thank the authors for their efforts. Additional results have been added to strengthen their conclusions.

There are still some minor errors to be corrected before publication:

Supplementary Figure 9C: it looks like there is only the result for the control cell and not the one for the hybrid cell as stated in the caption. Could you please check along with the color of the curves.

Supplementary Figure 11: could the authors mention in the caption the operating conditions?

In page 7, the added text refers to Supplementary Figure 10 but it seems that it has to refer to Supplementary Figure 9. Could the authors check?

Page 7, line 300: "Meanwhile, C are proton conductive". Typo error?

Arnaud Morin

Reviewer #2 (Remarks to the Author):

The authors have provided a robust response to reviewers' comments and the revised manuscript is acceptable for publication subject to resolution of some minor issues which have become apparent in the revised version:

In the first paragraph of Section 1 of the text, it is stated that the 'reaction was conducted using sodium tungstate and ammonium sulfide as precursors'; ammonium sulfate is indicated in the supplementary document.

In the last paragraph of Section 3 of the text, the statement 'Meanwhile, C are proton conductive' needs clarification, and

Supplementary Figure numbers subsequent to Supplementary Figure 9 in the text require to be corrected

Response to Reviewers

Response to Reviewer 1:

Q1: Supplementary Figure 9C: it looks like there is only the result for the control cell and not the one for the hybrid cell as stated in the caption. Could you please check along with the color of the curves?

A1: Thank you for your question. Figure 9C illustrates the power difference between the hybrid cell and the control cell during the transient operation, so there is only one curve shown in the figure. We changed the color of the curve to pink to avoid confusion.

Q2. Supplementary Figure 11: could the authors mentioned in the caption the operating conditions?

A2: Thank you for the reminder. The operation condition was added to the end of the caption. (All the steady-state performance of the hybrid cell and the control cell was tested at 50 °C, 100 % humidified H₂ (stoichiometry = 1.5) and O₂ (stoichiometry = 4) were fed to the anodes and cathodes, respectively.)

Q3. In page 7, the added text refers to Supplementary Figure 10 but it seems that it has to refer to Supplementary Figure 9. Could the authors check?

A3: Thank you for carefully checking the manuscript. The reviewer is correct. The cited figure should be Supplementary Figure 9. We have checked through the manuscript and corrected all the citation of Supplementary Figures.

Q4. Page 7, line 300: "Meanwhile, C are proton conductive". Typo error?

A4: Thank you for pointing out the mistake. It is a typo. "C" should be "H_xWO₃".

Response to Reviewer 2:

Q1: In the first paragraph of Section 1 of the text, it is stated that the 'reaction was conducted using sodium tungstate and ammonium sulfide as precursors'; ammonium sulfate is indicated in the supplementary document.

A1: Thank you for carefully check. It should be ammonium sulfate. The typo in the main text was corrected.

Q2: In the last paragraph of Section 3 of the text, the statement 'Meanwhile, C are proton conductive' needs clarification, and Supplementary Figure numbers subsequent to Supplementary Figure 9 in the text require to be corrected

A2: Thank you for pointing out the mistakes. It is a typo in the last paragraph of Section 3 of the text. "C" should be " H_xWO_3 ". Supplementary Figure numbers subsequent to Supplementary Figure 9 in the text were all corrected.